thermodynamics/artificial intelligence

compressor performance modelling, characteristic map, support vector machine nonlinear regression, neural network

**Author for correspondence:**
Siyu Xu
e-mail: xsy19970128@163.com

# Compressor performance modelling method based on support vector machine nonlinear regression algorithm

Yulong Ying[1], Siyu Xu[1], Jingchao Li[2] and Bin Zhang[3]

[1]School of Energy and Mechanical Engineering, Shanghai University of Electric Power, Shanghai 200090, People's Republic of China
[2]School of Electronic and Information, Shanghai Dianji University, Shanghai, People's Republic of China
[3]Department of Mechanical Engineering, Kanagawa University, Yokohama, Japan

YY, 0000-0002-3867-5893; SX, 0000-0002-3705-7620; JL, 0000-0003-4338-7084

To overcome the difficulty of having only part of compressor characteristic maps including on-design operating point, and accurately calculate compressor thermodynamic performance under variable working conditions, this paper proposes a novel compressor performance modelling method based on support vector machine nonlinear regression algorithm. It is compared with the other three neural network algorithms (i.e. back propagation (BP), radial basis function (RBF) and Elman neural networks) from the perspective of interpolation and extrapolation accuracy as well as calculation time, to prove the validity of the proposed method. Application analyses indicate that the proposed method has better interpolation and extrapolation performance than the other three neural networks. In terms of flow characteristic map representation, the root mean square error (RMSE) of the extrapolation performance at higher and lower speed operating area by the proposed method is 0.89% and 2.57%, respectively. And the total RMSE by the proposed method is 2.72%, which is more accurate by 47% than the Elman algorithm. For efficiency characteristic map representation, the RMSE of the extrapolation performance at higher and lower speed operating area by the proposed method is 2.85% and 1.22%, respectively. And the total RMSE by the proposed method is 1.81%, which is more accurate by 35% than the BP algorithm. Moreover, the proposed method has better real-time performance compared with the other three neural network algorithms.

**Figure 1.** Compressor characteristic maps. (*a*) Compressor flow characteristic map. (*b*) Compressor efficiency characteristic map.

## 1. Introduction

The gas turbine is an internal combustion type power machine which uses a continuously flowing gas as working medium to drive an impeller to rotate at high speed and converts thermic energy into mechanical work. It has been widely used in power plant, ship industry and transportation industry [1,2] due to the excellent performance such as flexible start-up, high thermal efficiency, short construction period and low pollution emission. In the process of gas turbine operation, the main components (such as turbines, compressors and combustion chambers) may be affected by the harsh working conditions inside the engine and the surrounding environmental pollution [3,4]. As the operating time increases, component performance degradation or damage may occur, resulting in various serious faults [5]. Moreover, when the performance degradation or damage of these components happens, the intrinsic performance parameters of these components will also vary, resulting in changes of gas path measurable parameters [6]. Therefore, accurate gas turbine thermodynamic modelling plays a key role in gas turbine performance analysis and gas path diagnosis. The veracity of a gas turbine performance model is chiefly determined by the expression precision of its component characteristic maps, especially for compressor maps.

The compressor performance expressed in Cartesian coordinates is usually regarded as compressor characteristic maps, as shown in figure 1, and the compressor characteristic maps are further used in the thermodynamic model of gas turbine engines [7]. Compressor characteristic maps (i.e. the flow characteristic map and efficiency characteristic map) can be described by four absolute parameters (i.e. mass flow rate, pressure ratio, isentropic efficiency and rotational speed) or four similar corrected parameters [8]. Among these four parameters characterizing the operating point of the compressor, as long as any two of them have been determined, the other two parameters are determined accordingly. One of the toughest problems in the development of gas turbine performance model is compressor modelling [9]. Usually, only part of the compressor characteristic maps containing the on-design point is obtained by an engine test [10] bench under different operating conditions or by numerical simulation over computational fluid dynamics with high cost [11].

A lot of efficacious methods have been proposed to improve the calculation precision of the thermodynamic performance model for gas turbine engines, mainly through correcting the known component characteristic maps [12–14] or producing new ones [15] based on gas path measurable parameters. Simani *et al.* [16] introduced a gas turbine thermodynamic modelling method which used an optimization algorithm to seek an optimal set of scaling factors for the component maps, and subsequently the proposed method was expanded by Lambiris *et al.* [17]. Kong *et al.* [18] presented a method based on the existing component characteristic maps and the scaling factors obtained under the on-design operating point, to acquire new component characteristic maps under variable working conditions over system identification. Recently, the artificial neural networks (ANNs) have been theoretically used to close in upon any nonlinear function through a proper network structure, due to the high degree of nonlinear mapping capability [19–23]. For expressing compressor characteristic maps mathematically, Yu *et al.* [24] developed a data-based tri-layer back propagation (BP) neural network applied to the Levenberg–Marquardt algorithm. Ghorbanian & Gholamrezaei [25] investigated the modelling capability of various neural networks, as well as introducing the two models in the performance map simulation. Peng *et al.* presented a method for simulating compressor characteristics based on the radial basis function (RBF) neural network. A hybrid ANNs method integrated with partial least square was used to model thermodynamic performance for a scroll compressor by Tian *et al.* [26].

Although, many efficient methods have been performed to generate and predict the compressor characteristic maps in unknown regions based on the known and limited experimental data from the manufacturers or performance decks, the simulation accuracy of the conventional methods for the interpolation and extrapolation performance of compressor characteristic maps under off-design working conditions is usually unsatisfactory. Due to the high nonlinearity of these characteristic parameters, it is tough to establish a mathematical expression via a small amount of sample. Moreover, the training process of ANNs, which is similar to a black box problem, is very complicated and time-consuming, and it is difficult to correctly reflect the concrete input–output relationships of a compressor system in many cases. The networks are very susceptible to the network weights of initialization and are prone to converge to diverse local minima with different weights, failing network training. Therefore, it is necessary to propose a compressor characteristic map expression method with perfect interpolation and extrapolation performance to accurately realize the thermodynamic calculation of the compressor performance under off-design working conditions. In order to further improve the accuracy of the compressor performance model in a small amount of experimental data, this paper puts forward a novel method for representing the compressor characteristic maps based on support vector machine (SVM) nonlinear regression algorithm. Furthermore, the highlights of the paper are as follows:

(a) A novel method based on support vector machine nonlinear regression algorithm is proposed for representing the characteristic maps to accurately realize the thermodynamic calculation of compressor performance under variable working conditions.
(b) The accuracy of the interpolation and extrapolation performance by the proposed method and three other neural networks (BP, RBF and Elman) is comparatively analysed.
(c) The real-time performance by the proposed method and three other neural networks (BP, RBF and Elman) is also comparatively analysed.

The rest of the article is arranged as below. In §2, the detail description of the algorithms is developed. Furthermore, §3 presents the comparison and analysis of the various methods, with the conclusions in §4.

# 2. Methodology

## 2.1. Traditional methods

### 2.1.1. BP neural network

BP algorithm is an effective multi-layer feed-forward network, whose mathematical principle is the forward propagation of the signal and the back propagation of the error. The ultimate network outputs are as close as possible to the expected outputs by continuously regulating the network connection weights by network training. The network structure is constituted of the input layer, hidden layer and output layer, as shown in figure 2. $w_{ij}$ is the weight from the input layer to the hidden layer, and $w_{jk}$ denotes the weight from the hidden layer to the output layer, where $i$, $j$ and $k$ represents the number of neurons in each layer, respectively.

It is indispensable to initialize the following parameters for a BP neural network before training.

(1) The number of neurons on the hidden layer

The number of neurons on the hidden layer determines the expressive power of the network, which determines the complicated degree of the decision boundary. When the number of neurons on the hidden layer is large, the training error may become small, but the error rate of the test sample will be high in this case, resulting in an over-fitting phenomenon. On the other hand, if the number of neurons on the hidden layer is tiny, the network does not have enough freedom to fit the training set, which also leads to a high test error rate.

(2) Activation function

The activation function from the input layer to the hidden layer generally requires nonlinearity, saturation and continuity, while the activation function from the hidden layer to the output layer is linear. For the compressor characteristic parameters, the input and output parameters have a highly nonlinear relationship, and therefore the sigmoid transfer function is adopted in the hidden layer. There are two types of sigmoid transfer functions that are frequently used as follows.

For the logarithmic sigmoid transfer function, the function expression in Matlab toolbox is logsig as follows:

$$f = \frac{1}{1 + e^{-n}}. \tag{2.1}$$

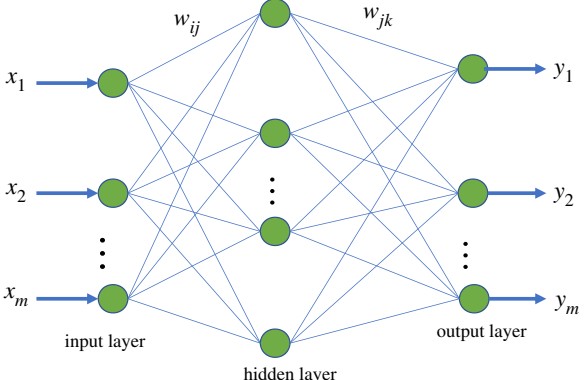

**Figure 2.** Schematic diagram of a BP neural network.

For the hyperbolic tangent sigmoid, the function expression in Matlab toolbox is tansig as follows:

$$f = \frac{e^n - e^{-n}}{e^n + e^{-n}}. \tag{2.2}$$

(3) Initialize weights

The weights cannot be simply initialized to 0, otherwise the learning process will not start. The weights from the input layer should be within the range of $-(1/\sqrt{D}) < w_{ij} < (1/\sqrt{D})$, where $D$ is the dimensionality of the input vector. The weights from the hidden layer to the output layer should be within the range of $-(1/\sqrt{l}) < w_{jk} < (1/\sqrt{l})$, where $l$ is the number of neurons on the hidden layer.

(4) Learning rate and threshold

The learning rate determines the network training convergence speed, which can be set to 0.1 initially. If the training speed is too slow, the learning rate can be increased. If the criterion function is diverged during the learning process, the learning rate will be reduced. The threshold determines whether the training process can be stopped.

### 2.1.2. RBF neural network

Both the BP neural network and the RBF neural network are nonlinear multi-layer feed-forward networks. For any BP network, there is always an RBF network that can replace it and vice versa. However, there are many differences between these two neural networks, such as network structure, training process and approximation performance.

The input layer of an RBF neural network is directly connected to the hidden layer, which is equivalent to transporting the input vector to the hidden layer directly. There are several activation functions for the hidden layer. Moreover, the most frequently applied activation function is the Gaussian function as follows:

$$\phi(r) = \exp\left(-\frac{r^2}{2\sigma^2}\right), \tag{2.3}$$

where $\sigma$ is an extended constant of the radial basis function, which reflects the width of the function image, and the smaller the $\sigma$, the narrower the width. The transfer function of the RBF neural network takes the distance $\|X - C_j\|$ from the input vector to the centre vector as an independent variable and replaces $r$ of the Gaussian function with $\|X - C_j\|$.

The parameters associated with the RBF neural network that need to be determined, are the data centre $C$, extended constant $\sigma$, and weights from the hidden layer to the output layer. The RBF neural network can determine the corresponding network topology according to specific problems, owing to the characteristics of self-learning, self-organization and adaptive functions. The structural diagram of an RBF neural network is shown in figure 3, where $y$ represents the actual output on the output layer, $P$ delegates the number of neurons on the hidden layer, $w$ denotes the weight from the hidden layer to the output layer and $\phi$ indicates the data centre of neurons on the hidden layer.

### 2.1.3. Elman neural network

The Elman neural network was put forward by J. L. Elman, aiming at the speech processing problem, which is a quintessential dynamic recursive neural network. A support layer is added to the hidden

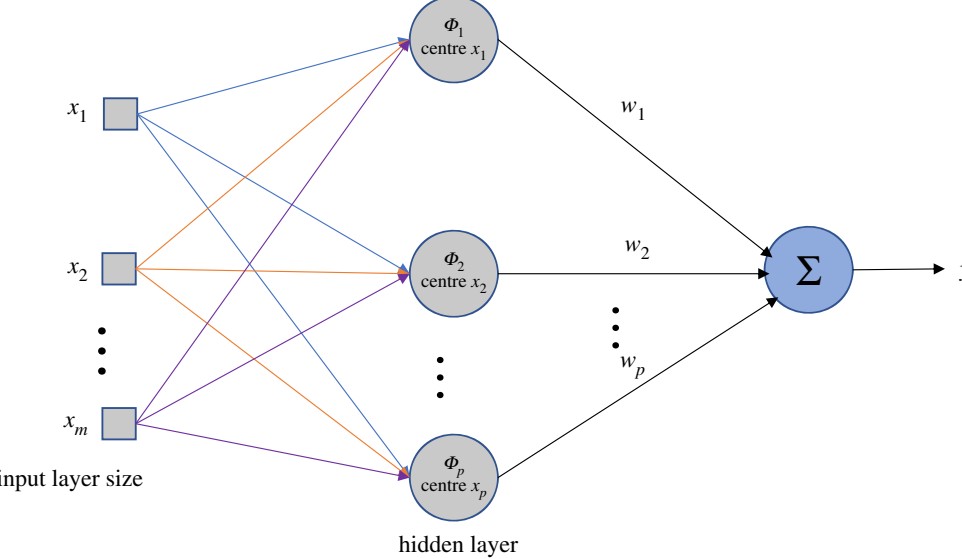

**Figure 3.** Structural diagram of an RBF neural network.

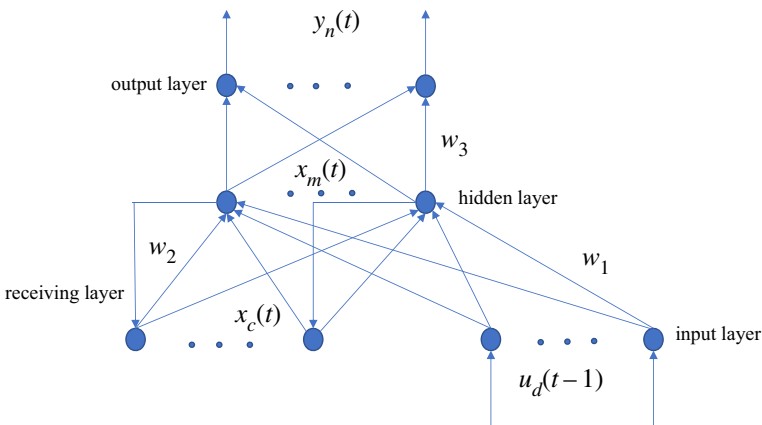

**Figure 4.** Structural diagram of an Elman neural network.

layer on the basis of a BP network structure to attain the goal of memory, such that the system could adapt to time-varying property and enhance the global stability of the network.

It has more computing power than the feed-forward network and can be applied to address the problem of fast optimization as well. The structural diagram of an Elman neural network is shown in figure 4.

The mathematical expressions of the Elman neural network are as follows:

$$y_n(t) = f(w_3 x_m(t)), \tag{2.4}$$

$$x_m(t) = g(w_2 x_c(t) + w_1(u_d(t-1))) \tag{2.5}$$

and

$$x_c(t) = x_m(t-1), \tag{2.6}$$

where $y_n$ is the $n$-dimensional output node unit vector; $x_m$ represents an $m$-dimensional middle layer node vector; $u_d$ indicates a $d$-dimensional input node vector; $x_c$ is an $m$-dimensional vector of the feedback state; $w_3$ means the weight from the intermediate layer to the output layer; $w_2$ means the weight from the connection layer to the intermediate layer; $w_1$ means the weight from the input layer to the intermediate layer; $f()$ is the transfer function of neurons on the output layer, which is the linear combination of the intermediate layer output; $g()$ is the transfer function of neurons on the intermediate layer, and the sigmoid transfer function is often used.

## 2.2. Support vector machine nonlinear regression algorithm

Support vector machine (SVM) has been applied extensively in data classification, regression estimation, function approximation and other fields. Its idea is very similar to that in the classification when SVM is applicable to the function regression. Therefore, only the concept of loss function needs to be introduced, and a $\varepsilon$-insensitive loss function proposed by Vapnik is represented by

$$L_\varepsilon = (h(x),y) = \begin{cases} 0, |h(x) - y| < \varepsilon \\ |h(x) - y| - \varepsilon \quad \text{others} \end{cases} \tag{2.7}$$

where $\varepsilon$, the insensitive coefficient, is applied to determine the fitting accuracy.

The extension from linear SVM to nonlinear SVM is by means of a kernel function, whose basic principle is to transform the input to another space by mapping and use a linear SVM in the new space. Furthermore, the radial basis kernel, a commonly used kernel function, is a kind of mapping, as shown in the following equation:

$$k(x,x_i) = \exp\left\{ -\frac{\|x - x_i\|^2}{\sigma^2} \right\}. \tag{2.8}$$

Assume that the nonlinear model is expressed as

$$\hat{h}(x,w) = (w \cdot \varphi(x)) + b. \tag{2.9}$$

When a dataset $\{x_i, y_i\}, i = 1, \ldots, n, x_i \in R^d, y_i \in R$ is fitted by the function, the unstructured deviation of the whole training sample is $\varepsilon$ as follows:

$$\left. \begin{aligned} y_i - ((w \cdot \varphi(x_i)) + b) \leq \varepsilon \\ ((w \cdot \varphi(x_i)) + b) - y_i \leq \varepsilon \end{aligned} \quad i = 1, \ldots, n. \right\} \tag{2.10}$$

A relaxation factor $\zeta_i \geq 0, \zeta_i^* \geq 0$ can be introduced in the case of considering the fitting error to make $1/2\|w\|^2$ the smallest according to the minimization criterion of structural risk.

$$\left. \begin{aligned} y_i - ((w \cdot \varphi(x_i)) + b) \leq \varepsilon + \zeta_i \\ ((w \cdot \varphi(x_i)) + b) - y_i \leq \varepsilon + \zeta_i^* \end{aligned} \quad i = 1, \ldots, n. \right\} \tag{2.11}$$

The optimization goal is correspondingly minimized.

$$J = \frac{1}{2}\|w\|^2 + C\sum_{i=1}^{n} (\zeta_i + \zeta_i^*), \tag{2.12}$$

where $C$ is the balance factor higher than naught.

Moreover, the standard $\varepsilon$ insensitive support vector regression machine is given as

$$\left. \begin{aligned} &\min \frac{1}{2}\|w\|^2 + C\sum_{i=1}^{n} (\zeta_i + \zeta_i^*) \\ &\text{subject to} \quad \begin{cases} y_i - ((w \cdot \varphi(x_i)) + b) \leq \varepsilon + \zeta_i \\ ((w \cdot \varphi(x_i)) + b) - y_i \leq \varepsilon + \zeta_i \\ \zeta_i \cdot \zeta_i^* \geq 0. \end{cases} \end{aligned} \right\} \tag{2.13}$$

The quadratic programming problem can be obtained using the same optimization method, and the Lagrange Equation is established as follows:

$$\begin{aligned} l(w,b,\zeta_i,\zeta_i^*) = &\frac{1}{2}\|w\|^2 + C\sum_{i=1}^{n} (\zeta_i + \zeta_i^*) - \sum_{i=1}^{n} \beta_i(\varepsilon + \zeta_i - y_i + (w \cdot \varphi(x_i)) + b) \\ &- \sum_{i=1}^{n} \beta_i^*(\varepsilon + \zeta_i^* + y_i - (w \cdot \varphi(x_i)) - b) - \sum_{i=1}^{n} (\eta_i\zeta_i + \eta_i^*\zeta_i^*). \end{aligned} \tag{2.14}$$

The partial derivatives of the above parameters (i.e. $w$, $b$, $\zeta_i$, and $\zeta_i^*$) should be equal to zero, then the following equation (2.15) can be obtained:

$$\left.\begin{aligned}
\frac{\partial l}{\partial w} &= w - \sum_{i=1}^{n} (\beta_i - \beta_i^*)\varphi(x_i) = 0, \\[4pt]
\frac{\partial l}{\partial b} &= \sum_{i=1}^{n} (\beta_i - \beta_i^*) = 0, \\[4pt]
\frac{\partial l}{\partial \zeta_i} &= C - \beta_i - \eta_i = 0 \\[4pt]
\frac{\partial l}{\partial \zeta_i^2} &= C - \beta_i^* - \eta_i^* = 0.
\end{aligned}\right\} \qquad (2.15)$$

and

Equation (2.12) is further written as

$$\left.\begin{aligned}
\min \frac{1}{2}\sum_{i,j=1}^{n} &(\beta_i - \beta_i^*)(\beta_j - \beta_j^*)\langle \varphi(x_i), \varphi(x_j)\rangle + \sum_{i=1}^{n} \beta_i(\varepsilon - y_i) + \sum_{i=1}^{n} \beta_i^*(\varepsilon + y_i) \\[4pt]
\text{subject to} \quad &\begin{cases} \sum_{i=1}^{n} (\beta_i - \beta_i^*) = 0 \\ \beta_i, \beta_i^* \in [0, C], \end{cases}
\end{aligned}\right\} \qquad (2.16)$$

where $\beta_i^*$ is the Lagrange multiplier, and the quadratic programming is solved.

$$w = \sum_{i=1}^{n} (\beta_i - \beta_i^*)\varphi(x_i). \qquad (2.17)$$

The $w$ cannot be expressed explicitly due to the facts that the nonlinear function $\varphi$ is unknown and the dimensionality of feature space is high enough. The task of the SVM algorithm is to introduce the kernel function technique so that the function regression bypasses the feature space and directly finds on the input space, avoiding the calculation of the nonlinear mapping $\varphi$.

Suppose the kernel function $k(x, x')$ satisfies the following formula:

$$k(x, x') = \{\varphi(x), \varphi(x')\}. \qquad (2.18)$$

Equation (2.16) can be adjusted as follows.

$$\left.\begin{aligned}
\min \frac{1}{2}\sum_{i,j=1}^{n} &(\beta_i - \beta_i^*)(\beta_j - \beta_j^*)k(x_i, x_j) + \sum_{i=1}^{n} \beta_i(\varepsilon - y_i) + \sum_{i=1}^{n} \beta_i^*(\varepsilon + y_i) \\[4pt]
\text{subject to} \quad &\begin{cases} \sum_{i=1}^{n} (\beta_i - \beta_i^*) = 0 \\ \beta_i, \beta_i^* \in [0, C]. \end{cases}
\end{aligned}\right\} \qquad (2.19)$$

According to the KKT condition, the following equation (2.20) can be obtained:

$$\left.\begin{aligned}
\beta_i(\varepsilon + \zeta_i - y_i + (w \cdot \varphi(x_i)) + b) &= 0 \\
\beta_i^*(\varepsilon + \zeta_i^* + y_i - (w \cdot \varphi(x_i)) - b) &= 0.
\end{aligned}\right\} \qquad (2.20)$$

Threshold $b$ is calculated by equation (2.21)

$$\left.\begin{aligned}
b &= y_i - (w \cdot \varphi(x_i)) - \varepsilon, \quad \beta_i \in (0, C) \\
b &= y_i - (w \cdot \varphi(x_i)) - \varepsilon, \quad \beta_i^* \in (0, C).
\end{aligned}\right\} \qquad (2.21)$$

The sample $x_i$ on or outside the boundary of the dead zone is corresponding to $\beta_i \neq 0$ and $\beta_i^* \neq 0$, which is called a support vector.

Thus, the following equation (2.22) can be obtained:

$$\hat{h} = \sum_{i=1}^{n} (\beta_i - \beta_i^*)k(x_i, x) + b. \qquad (2.22)$$

The support vector machine is a learning process that converts the input sample into a high dimensional feature space using a nonlinear conversion determined by an inner product and finds the

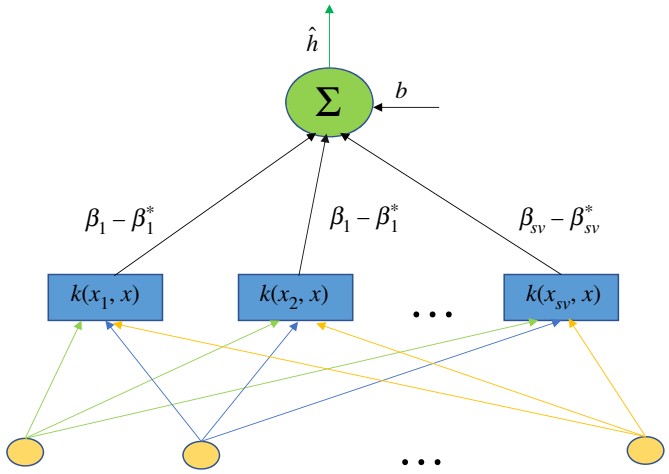

**Figure 5.** SVM regression diagram.

regression function in the new space. The output, a linear combination of the middle nodes, is similar to a neural network in form, and each node represents a particular support vector. Furthermore, its weight corresponds to the Lagrange multiplier as shown in figure 5, where $sv$ represents a support vector set.

The known compressor characteristic data is selected as a learning sample of the support vector machine model in this paper. Since SVM can only be applied for the approximation problem of single output function, the corresponding learning machine must be designed for each output to realize the model identification of multi-input and multi-output objects, as shown in figure 5.

For a compressor characteristic map, the pressure ratio and the rotational speed are considered as the input variables of a support vector machine model, and the mass flow rate and the isentropic efficiency are regarded as the output variables. The two support vector machines are connected in parallel to form training samples at different rotational speeds, and it should be emphasized that all available experimental data must be normalized prior to being trained.

The radial basis kernel function is exploited for a kernel function of the model, the width parameter as well as balance factor C are determined and optimized by a grid optimization programme, and the insensitive coefficient is found to be 0.01. The support vector machine model is obtained by training and analysing the compressor characteristic data at known rotational speeds. An inverse normalization process is performed to acquire a comparison between the training result and sample data after the model is output.

# 3. Application and analysis compressor component characteristics different algorithm prediction contrast

## 3.1. Forecasting comparison of the flow characteristics

In order to prove the rationality and effectivity of the proposed method, the other three neural network algorithms (i.e. BP, RBF and Elman neural networks) are compared from the perspective of interpolation and extrapolation accuracy as well as calculation time. The prediction of interpolation and extrapolation performance by the different algorithms for the flow characteristic map is as shown in figures 6–8.

To further compare the prediction accuracy of the above algorithms for the flow characteristic map, the root mean square error (RMSE) (as shown in equation (3.1)) between the experimental data and predicted data is introduced as an evaluation index of the model performance. Moreover, the comparative results are tabulated in table 1.

$$\text{RMSE} = \sqrt{\frac{\sum_{i=1}^{n}\left[(p_i - t_i)\right]^2}{n}},$$  (3.1)

where $p_i$ represents the $i$th prediction output, $t_i$ denotes the $i$th test sample data and $n$ is the sample size.

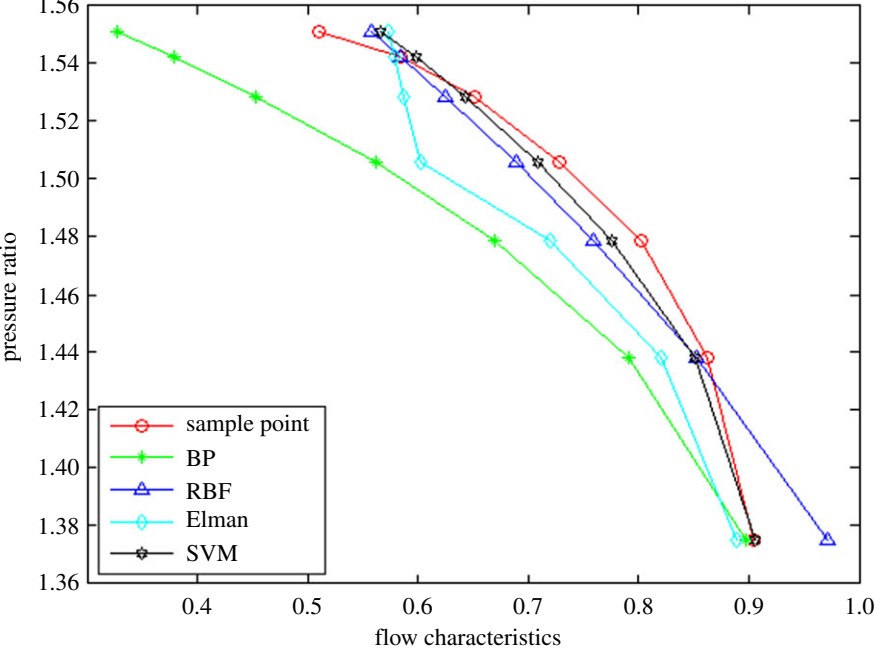

**Figure 6.** The comparison of extrapolation performance at lower speed operation area.

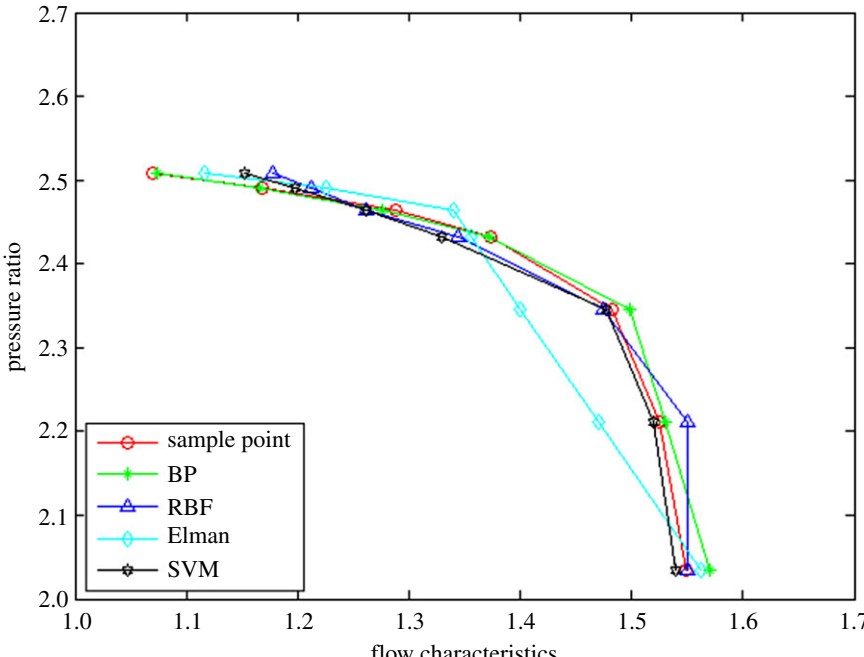

**Figure 7.** The comparison of interpolation performance.

Table 2 shows the time consumption comparison by different algorithms for predicting one operation point.

It can be observed from figures 6–8 and table 1 that the BP neural network algorithm has superior interpolation performance (IP) in comparison with the other three methods, but also the worst extrapolation (generalization) performance for the flow characteristic map. The RMSE of the extrapolation performance at higher (EPH) and lower (EPL) speed operating area by the proposed method is 0.89% and 2.57%, respectively. Moreover, the total RMSE by the proposed method is 2.72%, having an accuracy of 97%, which is more accurate by 47% than the Elman algorithm. In other words, the proposed method of SVM ameliorates the prediction performance of extrapolation and interpolation, the extrapolation performance at higher speed operating area in particular. The trend is

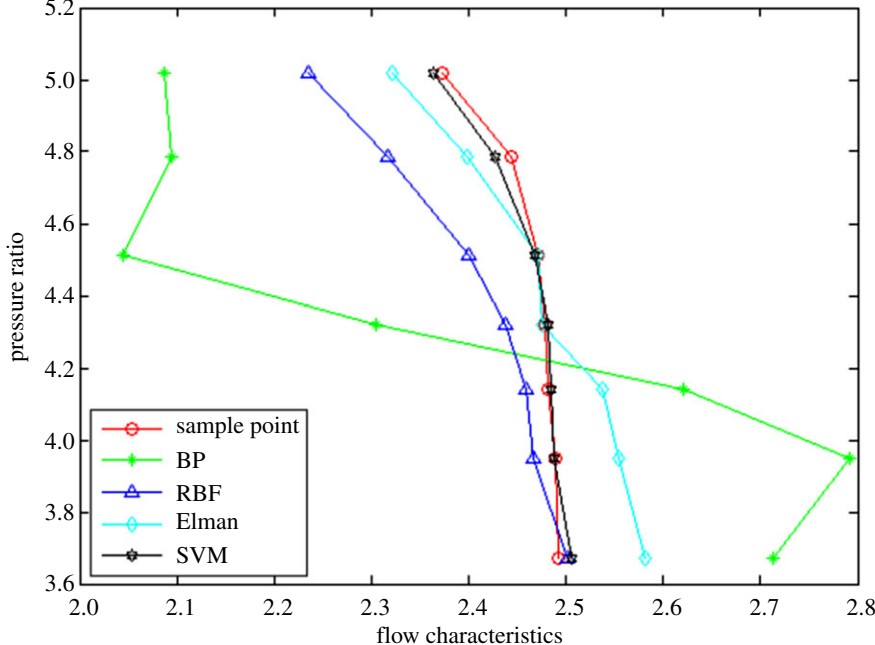

**Figure 8.** The comparison of extrapolation performance at higher speed operation area.

**Table 1.** The prediction accuracy comparison of extrapolation and interpolation performance by different methods for flow characteristic map.

| algorithms | RMSE | | | |
| --- | --- | --- | --- | --- |
| | EPL | IP | EPH | total |
| BP | 15.408864% | 1.1472180% | 28.653728% | 18.795259% |
| RBF | 3.9349806% | 4.7413000% | 7.8800000% | 5.7750000% |
| Elman | 5.6672845% | 5.1060174% | 5.3362097% | 5.3747763% |
| SVM | 2.5688705% | 3.8588333% | 0.8907238% | 2.7237937% |

**Table 2.** Time consumption comparison by different methods for predicting one operation point.

| algorithms | BP | RBF | Elman | SVM |
| --- | --- | --- | --- | --- |
| time consumption (s) | 0.084607 | 0.021256 | 0.029781 | 0.006427 |

in excellent agreement with the sample curve and conforms to the variation law of the actual compressor characteristic data. According to table 2, the time consumption by the proposed method for predicting one operation point is the least, which shows the approach based on the SVM algorithm has better real-time performance.

## 3.2. Forecasting comparison of the efficiency characteristics

The comparison of extrapolation and interpolation performance for the efficiency characteristic map based on the different algorithms is shown in figures 9–11.

Here, the standard presented in equation (3.1) is also applied to evaluate the prediction accuracy of extrapolation and interpolation performance. Moreover, table 3 shows comparative results.

Similarly, the time consumption comparison for predicting one operation point is listed in table 4.

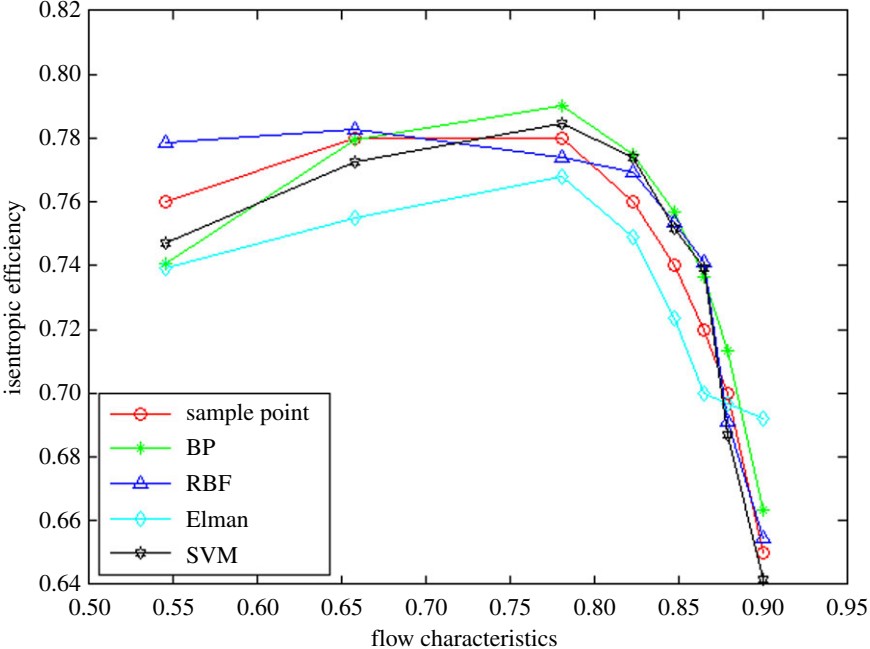

**Figure 9.** The comparison of extrapolation performance at lower speed operation area.

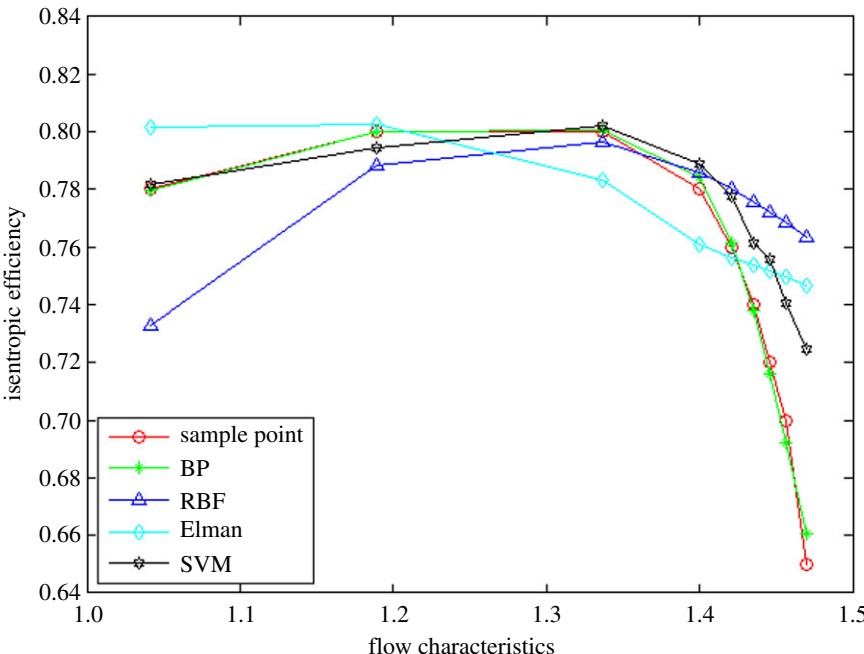

**Figure 10.** The comparison of interpolation performance.

It is evident from figures 9–11 and table 3 that the BP neural network algorithm has superior interpolation performance compared with the other three methods for the efficiency characteristic map, while it has perfect extrapolation (generalization) performance using the SVM regression algorithm. The RMSE of the extrapolation performance at higher and lower speed operating area by the proposed method is 2.85% and 1.22%, respectively. Moreover, the total RMSE by the proposed method is 1.81%, having an accuracy of 98%, which is more accurate by 35% than the BP algorithm. In other words, the proposed method ameliorates the prediction performance of extrapolation and interpolation, the extrapolation performance of higher speed operating area in particular. It can not only accurately reflect the changing trend of data, but also approximate the test data with a small

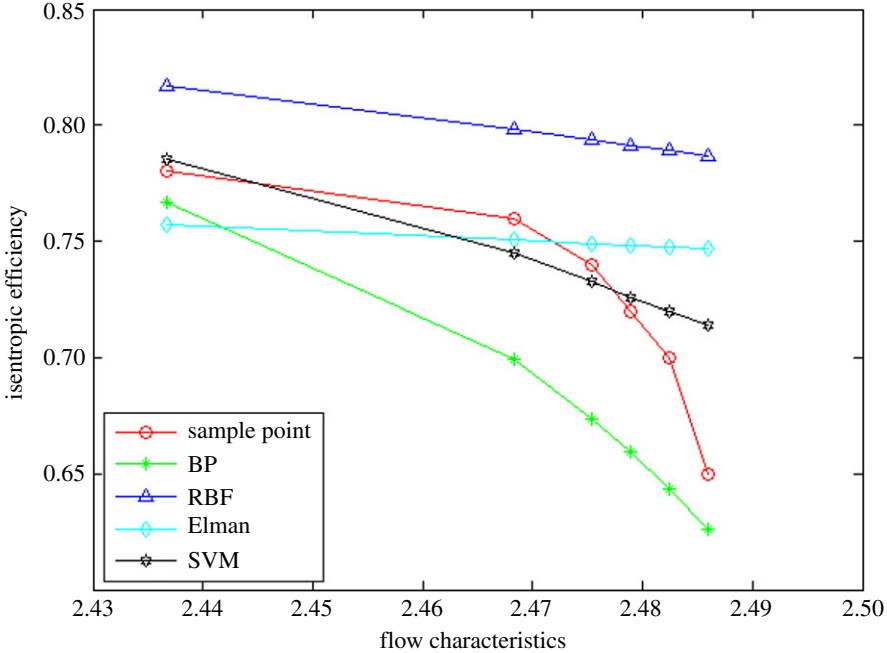

**Figure 11.** The comparison of extrapolation performance at higher speed operation area.

**Table 3.** The prediction accuracy comparison of extrapolation and interpolation performance by different methods for efficiency characteristic map.

| algorithms | RMSE | | | |
| --- | --- | --- | --- | --- |
| | EPL | IP | EPH | total |
| BP | 1.4182046% | 0.4797078% | 5.1203607% | 2.7620886% |
| RBF | 1.2306698% | 5.2004624% | 7.9100000% | 5.2366140% |
| Elman | 2.1793671% | 3.9644160% | 4.6770583% | 3.6753925% |
| SVM | 1.2189916% | 1.2678343% | 2.8497601% | 1.8067621% |

**Table 4.** Time consumption comparison by different methods for predicting one operation point.

| algorithms | BP | RBF | Elman | SVM |
| --- | --- | --- | --- | --- |
| time consumption (s) | 0.017188 | 0.029614 | 0.015301 | 0.005847 |

error. And from table 4, the time consumption by the proposed method for predicting one operation point is the least, which shows the approach based on the SVM algorithm has better real-time performance.

## 4. Conclusion and discussion

A novel method integrated with the support vector machine nonlinear regression algorithm is put forward for representing the characteristic maps in this article. The regression model based on the known compressor characteristic data is established in the Matlab operation as well as simulation environments, and the conclusions are as follows:

(1) The proposed method based on the SVM algorithm has superior overall interpolation and extrapolation performance than that of other commonly used neural network methods on the representation of a compressor characteristic map, especially for the extrapolation performance of

higher speed operating area, facilitating subsequent the calculation of compressor thermodynamic performance under variable working conditions.

(2) The time consumption by the proposed method for predicting one operation point is the least, which can ameliorate the real-time calculation capability of the dynamic performance simulation on a gas turbine engine.

(3) The introduction of the proposed method, enabling to address the problems of high-dimensional and machine learning in small sample data, as well as to avoid the structural selection of neural networks and sinking into local minima, could achieve the global optimization.

(4) The proposed method can be anticipated to ameliorate the existing component behaviour-based engine performance model for health monitoring, thermodynamic analysis, fault diagnosis and prognosis.

Data accessibility. The compressor characteristic lines for testing are from attachments of the General characteristics of the compressor test points data and General characteristics of the compressor prediction points data. The datasets supporting this article have been uploaded as part of the electronic supplementary material.

Authors' contributions. S.X. participated in the design of the study and drafted the manuscript; Y.Y. and J.L. participated in data analysis; B.Z. participated in the modification of the revised paper. All authors gave final approval for publication.

Competing interests. The authors declare that there is no conflict of interests regarding the publication of this paper.

Funding. The research of the paper is supported by the National Natural Science Foundation of China (nos. 51806135 and 61603239) and the State Key Lab of CEMEE Foundation (CEMEE2016K0102A).

Acknowledgement. This research is supported by the National Natural Science Foundation of China (nos. 51806135 and 61603239) and the State Key Lab of CEMEE Foundation (CEMEE2016K0102A). And the authors are grateful.

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
