## [Reviewer comments · Royal Society Open Science]

Review History

RSOS-191596.R0 (Original submission)

Review form: Reviewer 1

Is the manuscript scientifically sound in its present form?

Yes

Are the interpretations and conclusions justified by the results?

Yes

Is the language acceptable?

Yes

Do you have any ethical concerns with this paper?

No

Have you any concerns about statistical analyses in this paper?

No

Recommendation?

Accepted with minor revision (please list in comments)

Comments to the Author(s)

- 1) The abstract should be condensed and reorganized, The abstract should point out the contribution of the paper over some qualitative and quantitative results obtained by the proposed approach.
- 2) The quality of the table 2 could be improved, please edit it.
- 3) The written English should be modified carefully to avoid grammatical errors.

Review form: Reviewer 2**Is the manuscript scientifically sound in its present form?**

No

Are the interpretations and conclusions justified by the results?

Yes

Is the language acceptable?

Yes

Do you have any ethical concerns with this paper?

No

Have you any concerns about statistical analyses in this paper?

Yes

Recommendation?

Major revision is needed (please make suggestions in comments)

Comments to the Author(s)

Review Comments

Manuscript Number: RSOS-191596

Title: Compressor characteristic modeling method based on Support Vector Machine Regression Algorithm

In this work, author proposed a novel method for representing the compressor characteristic line based on support vector machine (SVM) nonlinear regression and provide the comparative analysis with three neural network algorithm, i.e., BP, RBF and Elman neural networks to verify and validate the performance in terms of perspective of interpolation and generalization accuracy and computational time. Results show that that the SVM regression algorithm has better interpolation and extrapolation performance than the other three neural networks. Additionally, the SVM regression method proposed in the article has better computational real-time performance while ensuring the required accuracy. Generally, topic is interesting and contribution has some merits but before publication in the journal I have few suggestions:

1. The title " Compressor characteristic modeling method based on Support Vector Machine Regression Algorithm " have no reflection of novelty and clarity. Well-known problem and its analysis with well-known method. Please revise if possible.
- 2 These sentences in abstract "The development of gas turbine industry is of essential strategic significance for promoting the adjustment, transformation and upgrading of national industrial structure and improving the quality and efficiency of economic growth. In the process of actual

thermodynamic modeling, only part of compressor characteristic lines containing design condition points can be obtained by test bed or flow analysis scheme. Therefore, it is necessary to propose a method for expressing the compressor characteristic map with perfect interpolation and generalization performance in order to accurately calculate the thermal calculation of the compressor under variable working conditions." should not be the part of abstract, it better suited in introduction section. While elaborative quantitative and qualitative advantages of the proposed procedure should be provided in the abstract of the manuscript.

3. Introduction section is too short and mainly based on old reference (Only two to three from last five years). However, introduction can be made appropriate by segmented in the introduction into three separate subsections, (1 Introduction, 1.1 related work, 1.2 Innovative contribution, 1.3 organization. Moreover, in the introduction section salient feature of the proposed methodology should be listed in bullet form.

4. Literature review regarding the journal applications of neural networks in diversified field in lacking in the introduction section. Authors are advice to see the recent paper of Prof. Dumitru Baleanu [r1-r3] and Prof A. M. Wazwaz [r4] and see how in different applications ANN is applied such as astrophysics, plasma physics, atomic physics, thermodynamics, electromagnetic, machines, nanotechnology, fluid mechanics, electrohydrodynamics, signal processing, power, energy, bioinformatics, economic and finance are provided there.

[r1] A new stochastic computing paradigm for the dynamics of nonlinear singular heat conduction model of the human head. The European Physical Journal Plus, 2018 133(9), p.364.

[r2] A new stochastic computing paradigm for nonlinear Painlevé II systems in applications of random matrix theory. 2018 The European Physical Journal Plus, 133(7), p.254.

[r3] Design of computational intelligent procedure for thermal analysis of porous fin model. 2019 Chinese Journal of Physics, 59, pp.641-655.

[r4] Neuro-heuristics for nonlinear singular Thomas-Fermi systems. Applied Soft Computing, 2018 65, pp.152-169.

5. Please provide a pseudocode of the proposed methodology in elaborative manner, i.e., statements and equations for input, output and intermediate steps.

6. Results and discussion section is too brief. Please include the elaborative description of the results and provide a comparative study on statistical analysis (T-Test, ANOVA etc) of the results.

7. Conclusion section the role of present study for future development of the research should be narrated.

Decision letter (RSOS-191596.R0)

30-Oct-2019

Dear Ms Xu,

The editors assigned to your paper ("Compressor characteristic modeling method based on Support Vector Machine Regression Algorithm") have now received comments from reviewers. We would like you to revise your paper in accordance with the referee and Associate Editor suggestions which can be found below (not including confidential reports to the Editor). Please note this decision does not guarantee eventual acceptance.

Please submit a copy of your revised paper before 22-Nov-2019. Please note that the revision deadline will expire at 00.00am on this date. If we do not hear from you within this time then it will be assumed that the paper has been withdrawn. In exceptional circumstances, extensions may be possible if agreed with the Editorial Office in advance. We do not allow multiple rounds

of revision so we urge you to make every effort to fully address all of the comments at this stage. If deemed necessary by the Editors, your manuscript will be sent back to one or more of the original reviewers for assessment. If the original reviewers are not available, we may invite new reviewers.

- Data accessibility

<http://datadryad.org/submit?journalID=RSOS&manu=RSOS-191596>

- Competing interests

- Authors' contributions

- Acknowledgements

- Funding statement

Kind regards,
Anita Kristiansen
Editorial Coordinator
Royal Society Open Science
openscience@royalsociety.org

on behalf of Dr Derek Abbott (Associate Editor) and R. Kerry Rowe (Subject Editor)
openscience@royalsociety.org

Reviewers' Comments to Author:

Reviewer: 1

Comments to the Author(s)

- 1) The abstract should be condensed and reorganized, The abstract should point out the contribution of the paper over some qualitative and quantitative results obtained by the proposed approach.
- 2) The quality of the table 2 could be improved, please edit it.
- 3) The written English should be modified carefully to avoid grammatical errors.

Reviewer: 2

Comments to the Author(s)

Review Comments

Manuscript Number: RSOS-191596

Title: Compressor characteristic modeling method based on Support Vector Machine Regression Algorithm

In this work, author proposed a novel method for representing the compressor characteristic line based on support vector machine (SVM) nonlinear regression and provide the comparative analysis with three neural network algorithm, i.e., BP, RBF and Elman neural networks to verify

and validate the performance in terms of perspective of interpolation and generalization accuracy and computational time. Results show that the SVM regression algorithm has better interpolation and extrapolation performance than the other three neural networks. Additionally, the SVM regression method proposed in the article has better computational real-time performance while ensuring the required accuracy. Generally, topic is interesting and contribution has some merits but before publication in the journal I have few suggestions:

1. The title " Compressor characteristic modeling method based on Support Vector Machine Regression Algorithm " have no reflection of novelty and clarity. Well-known problem and its analysis with well-known method. Please revise if possible.
- 2 These sentences in abstract "The development of gas turbine industry is of essential strategic significance for promoting the adjustment, transformation and upgrading of national industrial structure and improving the quality and efficiency of economic growth. In the process of actual thermodynamic modeling, only part of compressor characteristic lines containing design condition points can be obtained by test bed or flow analysis scheme. Therefore, it is necessary to propose a method for expressing the compressor characteristic map with prefect interpolation and generalization performance in order to accurately calculate the thermal calculation of the compressor under variable working conditions." should not be the part of abstract, it better suited in introduction section. While elaborative quantitative and qualitative advantages of the proposed procedure should be provided in the abstract of the manuscript.
3. Introduction section is too short and mainly based on old reference (Only two to three from last five years). However, introduction can be made appropriate by segmented in the introduction into three separate subsections, (1 Introduction, 1.1 related work, 1.2 Innovative contribution, 1.3 organization. Moreover, in the introduction section salient feature of the proposed methodology should be listed in bullet form.
4. Literature review regarding the journal applications of neural networks in diversified field in lacking in the introduction section. Authors are advice to see the recent paper of Prof. Dumitru Baleanu [r1-r3] and Prof A. M. Wazwaz [r4] and see how in different applications ANN is applied such as astrophysics, plasma physics, atomic physics, thermodynamics, electromagnetic, machines, nanotechnology, fluid mechanics, electrohydrodynamics, signal processing, power, energy, bioinformatics, economic and finance are provided there.
[r1] A new stochastic computing paradigm for the dynamics of nonlinear singular heat conduction model of the human head. The European Physical Journal Plus, 2018 133(9), p.364.
[r2] A new stochastic computing paradigm for nonlinear Painlevé II systems in applications of random matrix theory. 2018 The European Physical Journal Plus, 133(7), p.254.
[r3] Design of computational intelligent procedure for thermal analysis of porous fin model. 2019 Chinese Journal of Physics, 59, pp.641-655.
[r4] Neuro-heuristics for nonlinear singular Thomas-Fermi systems. Applied Soft Computing, 2018 65, pp.152-169.
5. Please provide a pseudocode of the proposed methodology in elaborative manner, i.e., statements and equations for input, output and intermediate steps.
6. Results and discussion section is too brief. Please include the elaborative description of the results and provide a comparative study on statistical analysis (T-Test, ANOVA etc) of the results.
7. Conclusion section the role of present study for future development of the research should be narrated.

Author's Response to Decision Letter for (RSOS-191596.R0)

See Appendix A.

Decision letter (RSOS-191596.R1)

29-Nov-2019

Dear Ms Xu,

It is a pleasure to accept your manuscript entitled "Compressor performance modeling method based on support vector machine nonlinear regression algorithm" in its current form for publication in Royal Society Open Science.

Kind regards,

Lianne Parkhouse
Royal Society Open Science
openscience@royalsociety.org

on behalf of Dr Derek Abbott (Associate Editor) and R. Kerry Rowe (Subject Editor)
openscience@royalsociety.org

Appendix A

Original Manuscript ID: RSOS-191596

Original Article Title: “Compressor characteristic modeling method based on Support Vector Machine Regression Algorithm”

To: Journal Editor

Re: Response to reviewers

Dear Editor,

Thank you for allowing a resubmission of our manuscript, with an opportunity to address the reviewers' comments.

We are uploading (a) our point-by-point response to the comments (below) (Response to Referees), (b) an updated manuscript with red highlighting indicating changes, and (c) a clean updated manuscript without highlights (main document).

For Reviewer 1

Recommendation: Accept (minor edits)

Comments:

1) The abstract should be condensed and reorganized. The abstract should point out the contribution of the paper over some qualitative and quantitative results obtained by the proposed approach.

Answer: We have rewritten the abstract and added some simple analyses, which points out the contribution of the paper over some qualitative and quantitative results obtained by the proposed approach.
Thanks for your advice.

2) The quality of the table 2 could be improved, please edit it.

Answer: We have modified this part in the paper and made each table to keep a consistent number of significant digits.
Thanks for your advice.

3) The written English should be modified carefully to avoid grammatical errors.

Answer: We have modified basic grammatical errors in the paper with red highlighting indicating changes.
Thanks for your advice.

Reviewer 2

Recommendation: Accept (minor edits)

Comments:

Title: Compressor characteristic modeling method based on Support Vector Machine Regression Algorithm

In this work, author proposed a novel method for representing the compressor characteristic line based on support vector machine (SVM) nonlinear regression and provide the comparative analysis with three neural network algorithm, i.e., BP, RBF and Elman neural networks to verify and validate the performance in terms of perspective of interpolation and generalization accuracy and computational time. Results show that that the SVM regression algorithm has better interpolation and extrapolation performance than the other three neural networks. Additionally, the SVM regression method proposed in the article has better computational real-time performance while ensuring the required accuracy. Generally, topic is interesting and contribution has some merits but before publication in the journal I have few suggestions:

1. The title " Compressor characteristic modeling method based on Support Vector Machine Regression Algorithm " have no reflection of novelty and clarity. Well-known problem and its analysis with well-known method. Please revise if possible.

Answer: We have changed the original title to "Compressor performance modeling method based on support vector machine nonlinear regression algorithm" with a minor change after careful consideration, which is considered to be more in line with the content of the article.

Thanks for your advice.

2. These sentences in abstract "The development of gas turbine industry is of essential strategic significance for promoting the adjustment, transformation and upgrading of national industrial structure and improving the quality and efficiency of economic growth. In the process of actual thermodynamic modeling, only part of compressor characteristic lines containing design condition points can be obtained by test bed or flow analysis scheme. Therefore, it is necessary to propose a method for expressing the compressor characteristic map with perfect interpolation and generalization performance in order to accurately calculate the thermal calculation of the compressor under variable working conditions." should not be the part of abstract, it better suited in introduction section. While elaborative quantitative and qualitative advantages of the proposed procedure should be provided in the abstract of the manuscript.

Answer: We have moved the sentences in abstract to introduction section and provided elaborative quantitative and qualitative advantages of the proposed procedure in the abstract, such as the analyses of root mean square error values and calculation time.

Thanks for your advice.

3. Introduction section is too short and mainly based on old reference (Only two to three from last five years). However, introduction can be made appropriate by segmented in the introduction into three separate subsections, (1 Introduction, 1.1 related work, 1.2 Innovative contribution, 1.3 organization. Moreover, in the introduction section salient feature of the proposed methodology should be listed in bullet form.

Answer: We have modified the introduction section and added several new articles from last five years. The related works are recombined according to the suggestions. The shortcomings of the traditional methods in predicting the interpolation and extrapolation performance of compressor characteristic maps are explained, and the novelty and advantages of the proposed method are introduced in this paper, which is listed in bullet form.

Thanks for your advice.

4. Literature review regarding the journal applications of neural networks in diversified field in lacking in the introduction section. Authors are advice to see the recent paper of Prof. Dumitru Baleanu [r1-r3] and Prof A. M. Wazwaz [r4] and see how in different applications ANN is applied such as astrophysics, plasma physics, atomic physics, thermodynamics, electromagnetic, machines, nanotechnology, fluid mechanics, electrohydrodynamics, signal processing, power, energy, bioinformatics, economic and finance are provided there.

[r1] A new stochastic computing paradigm for the dynamics of nonlinear singular heat conduction model of the human head. The European Physical Journal Plus, 2018 133(9), p.364.

[r2] A new stochastic computing paradigm for nonlinear Painlevé II systems in applications of random matrix theory. 2018 The European Physical Journal Plus, 133(7), p.254.

[r3] Design of computational intelligent procedure for thermal analysis of porous fin model. 2019 Chinese Journal of Physics, 59, pp.641-655.

[r4] Neuro-heuristics for nonlinear singular Thomas-Fermi systems. Applied Soft Computing, 2018 65, pp.152-169.

Answer: We have read the journal applications of neural networks in diversified field and added the suggested literatures as well as new literature[19] in the introduction section.

Thanks for your advice.

5. Please provide a pseudocode of the proposed methodology in elaborative manner, i.e., statements and equations for input, output and intermediate steps.

Answer: The pseudocode of the proposed methodology as follows:

Input: the pressure ratio and the rotational speed in compressor characteristic parameters

Output: the mass flow rate and the isentropic efficiency in compressor characteristic parameters

Steps :

(1) All sample data must be normalized.

(2) Establishment of the SVM model.

1) The best width parameter (which is represented by the letter g in the program) and balance factor (which is represented by the letter c in the program) are determined by a grid optimization method.

① The value range of c and g is from -10 to 10, and the interval is 0.5.

② The cg is set to a zero matrix of 41×41 (41 is the dimension of c and g). The initialization of best c

and best g are zero. Set five-fold cross validation mode, and the error is Inf. Set tolerance of termination criterion is 0.0001.

- ③ Generate $cg(i,j)$ within the value rang using for loop($i=1:41, j=1:41$).
- ④ Substitute into SVM model for training and judge whether $cg(i,j)$ at this time meets the termination conditions.
- ⑤ If the ④ met, the cycle is terminated to obtain the best c and best g . Otherwise, step ③ continues.

- 2) The insensitive coefficient is 0.01.
 - (3) Simulation Prediction of the SVM algorithm.
 - (4) The model output data subjects to an inverse normalization process.
 - (5) The experimental results are compared and analyzed.
- Thanks for your advice.

6. Results and discussion section is too brief. Please include the elaborative description of the results and provide a comparative study on statistical analysis (T-Test, ANOVA etc) of the results.

Answer: We have added the elaborative description of the results and a comparative study on statistical analysis with red highlighting indicating changes, such as the analyses of root mean square error values and calculation time.

Thanks for your advice.

7. Conclusion section the role of present study for future development of the research should be narrated.

Answer: We have modified the part in conclusion section and narrated the advantages of the present study for future development of the research.

Thanks for your advice.

Additional Statement:

In addition to addressing all of the reviewers' and editor's comments please also ensure that your revised manuscript contains the sections as appropriate before the reference list:

Answer: We have revised the manuscript and contained the required sections as appropriate before the reference list.

Thanks.

Best regards,

<Siyu Xu> et al.